# Cardiovascular Diseases and Pharmacomicrobiomics: A Perspective on Possible Treatment Relevance

**DOI:** 10.3390/biomedicines9101338

**Published:** 2021-09-28

**Authors:** Lavinia Curini, Amedeo Amedei

**Affiliations:** 1Department of Clinical and Experimental Medicine, University of Florence, 50139 Florence, Italy; lavinia.curini@unifi.it; 2SOD of Interdisciplinary Internal Medicine, Azienda Ospedaliera Universitaria Careggi (AOUC), 50139 Florence, Italy

**Keywords:** microbiome, heart disease, pharmacomicrobiomics, inflammation, immune response, personalized medicine

## Abstract

Cardiovascular diseases (CVDs), the most common cause of mortality in rich countries, include a wide variety of pathologies of the heart muscle and vascular system that compromise the proper functioning of the heart. Most of the risk factors for cardiovascular diseases are well-known: lipid disorders, high serum LDL cholesterol, hypertension, smoking, obesity, diabetes, male sex and physical inactivity. Currently, much evidence shows that: (i) the human microbiota plays a crucial role in maintaining the organism’s healthy status; and (ii) a link exists between microbiota and cardiovascular function that, if dysregulated, could potentially correlate with CVDs. This scenario led the scientific community to carefully analyze the role of the microbiota in response to drugs, considering this the right path to improve the effectiveness of disease treatment. In this review, we examine heart diseases and highlight how the microbiota actually plays a preponderant role in their development. Finally, we investigate pharmacomicrobiomics—a new interesting field—and the microbiota’s role in modulating the response to drugs, to improve their effectiveness by making their action targeted, focusing particular attention on cardiovascular diseases and on innovative potential treatments.

## 1. Introduction

Cardiovascular diseases (CVDs) are the leading cause of death worldwide and include a wide range of disorders, such as diseases of the heart muscle and vascular system [1].

Usually, CVDs are considered diseases of the Western countries, but recent evidence showed that the populations of emerging and even low-income countries also suffer from them. Despite advancements in primary and secondary CVD prevention, there are still significant disparities in cardiovascular healthcare across location and time.

Efforts to reduce this health disparity have been confirmed by a recent interest in developing new approaches to study the causes of risk factors, which include the social determinants of health [2]. Epidemiological research has mostly concentrated on identifying, altering and treating all the specific conditions statistically associated with the onset of cardiovascular disease; nevertheless, multiple cardiovascular risk factors are increasing at varying rates around the world [3].

The World Health Organization (WHO) reported in 2015 that cardiovascular diseases caused more than 17.7 million deaths worldwide, accounting for 31% of all global mortality [4,5,6].

Over recent years, it has been amply demonstrated that an incorrect diet, with a lack of fruit and vegetables but rich in red meats, and so in cholesterol, is one of the causes of intestinal disorders and infection which cause microorganism alterations and metabolic disorders, promoting the development of CVDs [7,8].

In fact, dietary carnitine (present predominantly in red meat) and lecithin have been shown to be metabolized by the gut microbiota (GM) to trimethylamine (TMA), which is metabolized by liver flavin monoxygenases to form trimethylamine-N-oxide (TMAO), an important gut microbe-dependent metabolite generated from dietary choline, betaine and L-carnitine [9,10].

Recently, the associations reported have focused attention on plasma TMAO levels as a potential determinant of cardiovascular diseases [11]. In particular, Seldin et al. investigated the effect of TMAO on endothelial and smooth muscle cell function in vivo. They reported that TMAO’s ability to stimulate inflammatory gene expression required the activation of nuclear factor-κB signaling [12]. In relation to this, a different study confirmed the role of TMAO in the activation of pro-inflammatory genes, inflammatory cytokines, adhesion molecules and chemokines and showed that an increase in the level of TMAO itself can induce the activation of the nuclear factor-κB pathway [13].

These findings suggest that TMAO can be considered a trigger for the early pathological process of CVDs by accelerating endothelial dysfunction, impacting on cell damage and inducing the oxidation of immune cells. 

TMAO could, therefore, be used as a circulating biomarker of cardiovascular risk and as a potential target for new therapeutic strategies based on the inhibition of various phases of its synthesis and on the monitoring of plasma levels. In fact, a prospective cohort study has shown that increased TMAO levels can be quantified and used as a predictor of adverse cardiovascular events, such as myocardial infarction (MI), stroke or aortic stenosis (AS) [14].

The human microbiota consists of the 10–100 trillion symbiotic microbial cells harbored by the human body, and the human microbiome consists of the genes that these cells contain. If we consider ourselves as a compound of microbial species and the human species, our genetic heritage is the result of the genes in our genome, and our metabolic characteristics are the sum of human and microbial traits.

Hence, a better understanding of the human microbiota could better explain the function and the dynamics of the resulting metabolites, holding great promise for predictive CVD biomarker discoveries and fine interventions [15,16,17,18].

At the same time, several studies have shed light on the metabolic inactivation of drugs affecting their therapeutic effect. Metabolic inactivation can be defined as an in vivo phenomenon that attenuates or inactivates the therapeutic effect of drugs, acting on the concentrations of numerous biotransformation activities that can directly influence their efficacy and toxicity. Drug metabolism can be influenced by a variety of factors in the gut, including host genotype, metabolic type, gut transit time, food intake and absorption, though reduction and hydrolysis reactions are the most commonly responsible for this phenomenon [19,20].

Linked to this, the study of pharmacomicrobiomics, a new, interesting branch, can be very useful for investigating how and why the effect of drugs can often be modulated by the microbiome and microbiota. Pharmacomicrobiomics investigations are utilized in the development of personalized medicine in order to reduce adverse drug reactions by giving the right drug to the right patient and at the right time. This phenomenon is characterized by tailoring medication therapy to the genetic architecture of the individual [21].

Therefore, the microbiota evaluation associated with pharmacomicrobiomics lends itself to relevant applications to develop new, specific strategies for microbiota manipulation in order to find targeted and personalized treatments for patients. The objective of investigating host–microbiota interactions by modeling them at different levels, starting from individual biochemical interactions—considering the patient’s health status, including diet and metabolism and, therefore, all those factors capable of predicting the modulation of the drug and, thus, its effectiveness—is certainly the right way to increase the effectiveness of administration with positive effects on patients. In this way, pharmacomicrobiomics could pave the way for innovative therapeutic approaches for a variety of diseases, including cardiovascular diseases.

## 2. The Most Representative Cardiovascular Diseases

As previously reported, cardiovascular diseases are the major cause of morbidity and mortality in developed countries, being the leading cause of death worldwide [22].

Commonly, the most frequent pathology—also defined as underlying disease—has been identified as atheromatous vascular disease. In fact, it results in coronary artery disease (CAD), peripheral vascular disease, cerebrovascular disease and the subsequent development of arrhythmias and heart failure [23].

The major risk factors are well known and can be summarized as hypertension, diabetes, high cholesterol levels, obesity and smoking [24,25]. Furthermore, there is substantial evidence that low cardiorespiratory fitness can be considered a predictor of cardiovascular and metabolic diseases [26]. 

Atherosclerosis, which is a chronic inflammatory condition, is the dominant cause of CVD, including a large range of diseases, such as stroke, myocardial infarction (MI) and heart failure.

Atherosclerosis mainly occurs in the intima of the vessels, influencing the normal blood flow and activating the endothelium with expression of adhesion molecules; the main direct CVD cause seems to be rupture of atherosclerotic plaques [27].

The deposition of microscopic cholesterol crystals in the intima and its underlying smooth muscle is the first step in the formation of these plaques. The plaques then expand inside the arteries due to the growth of fibrous tissues and surrounding smooth muscle, causing blood flow to be reduced. Sclerosis is caused by fibroblasts producing connective tissue and calcium deposits in the lesion [2,28,29].

Another important and frequent heart valve disorder is calcific aortic valve disease (CAVD), a dynamic process characterized by several steps, from valve mineralization to narrowing, with a consequent impairment of blood flow and calcium deposition [30]. Due to time-dependent wear and tear of the valve leaflets, as well as passive calcium deposition, this process was formerly regarded to be “degenerative”. Subsequent studies have instead shown that the presence of osteoblasts in vascular lesions suggests that valve calcification is a dynamic process that involves lipoprotein deposition and the presence of persistent inflammation [31]. CAVD and atherosclerosis share the same triggering event, for example, an increase in mechanical stress, altered shear stress and endothelial damage, which allows lipid infiltration and recruitment of inflammatory cells. These cells interact in processes that release reactive oxygen species (ROS), causing LDL oxidation [32].

The important differences between the two pathologies appear in the advanced disease stages, where smooth muscle cells are the active protagonists of chronic inflammation within the atherosclerotic plaque, and fibroblasts are the main cells involved in the valve mineralization process [30]. 

The hallmark of the early stages of CAVD is inflammation, characterized by the activation of valve endothelium, the expression of cell adhesion molecules (VCAM-1, ICAM-1) and, finally, the recruitment of immune cells [33,34].

The onset of the disease involves several events, such as the activation of valve interstitial cells (VICs) and sclerosis of the valve leaflets. The first CAVD stage is called aortic valve sclerosis, characterized by microcalcifications and valve thickening [35]. The final stage of disease is characterized by calcific aortic stenosis with calcified noduli on the valve leaflet surface, which hinder its mobility [36]. 

Moreover, the leaflets appear to be infiltrated by immune cells, with a large deposition of lipids, proteoglycans and cell debris [37]. The final result is an increased stiffness and the obstruction of the blood flow.

Another critical cardiovascular disease is endocarditis, an inflammatory condition of the endocardium that usually affects the heart valves. In the past several decades, the intervention of valve replacement and the use of cardiac devices such as permanent pacemakers have significantly contributed to the increase of infective endocarditis (IE) episodes [38]. 

IE is a life-threatening disease that has long-lasting effects in patients and most often occurs in left heart valves, in particular mitral or aortic valves [39]. 

IE disease is closely related to the virulence of the causative microorganism and the patient’s health status for triggering a good immune response. There are two main phenotypes of IE disease: (i) acute IE is a serious condition that develops over a period of days or weeks; and (ii) subacute IE follows a slower course, characterized by very slow progress over weeks or months [40,41]. Notably, valve disease (especially mitral valve disease) is considered a relevant predisposing factor of IE; nonetheless, nowadays the prevalence of IE is low, due to a decline in the prevalence of rheumatic heart disease [42].

Despite the new advances in prophylaxis, IE is one of the substantial causes of mortality in children and adolescents [43].

The only preventable cardiovascular disease so far is rheumatic heart disease (RHD), a chronic heart valve condition caused by an infection with the bacterium group A Streptococcus which can lead to acute rheumatic fever (ARF), a complication that leads to an autoimmune response. If Streptococcus A and ARF are left untreated, repeat infections are more likely to happen, which can cause permanent damage to heart valves. Long-term RHD consequences can include MI, stroke, atrial fibrillation (AT), endocarditis and aortic stenosis, which is the predominant valve lesion in RHD [44,45].

Although the global incidence of ARF and RHD has decreased, it remains endemic in low-income countries with gaps in healthcare and low availability of antibiotics; there these diseases are estimated to affect 20 million people and remain leading causes of cardiovascular death during the first five decades of life [45,46].

### The Role of Microbiota in Cardiovascular Diseases

Several pieces of scientific evidence have shown a tight correlation between infective agents and CVD pathogenesis and progression [47].

Many bacteria, such as *Chlamydia pneumoniae* (CP), *Chlamydia pneumoniae* (TWAR) and cytomegalovirus (CMV), have been detected in atheromatous plaques and associated with heart diseases [48]. 

In this regard, several studies have investigated the role of CP, examining how this bacterium can act as an inducer of atherosclerosis. In particular, by using immunohistochemistry techniques, PCR and tissue culture, it was revealed that CP increases lipid oxidation and modulates cell adhesion to the endothelial wall [49,50]. This bacterium can directly infect the cells involved in the process of atheroma formation, and it can be considered as a potential agent involved not only in coronary disease but also in myocardial vulnerability [51]. 

Another interesting study by Benagiano et al. investigated the cytokine and chemokine profile induced by *Chlamydophila pneumoniae* phospholipase D (CpPLD) infiltrating atherosclerotic lesions in patients with *C. pneumoniae* antibodies. The authors showed that plaque-derived T cells produced IL-17 in response to CpPLD; in addition, the CpPLD-specific CD4^+^ T lymphocytes carried out their helper function by monocyte matrix metalloproteinase and tissue factor production. Finally, CpPLD promoted Th17 cell migration and adhesion to endothelial cells. These results indicated that CpPLD drove the expression of different cytokines to induce a Th17 immune response that played a key role in the genesis of atherosclerosis [52].

Moreover, CP, together with *Helicobacter pylori*, has been associated with coronary heart disease (CHD), ischemic heart disease (IHD) and acute MI [49,53,54]. 

*Helicobacter pylori* infection, which usually promotes gastric disorders, especially peptic ulcer disease, has been detected in atheromatous plaques, where induced stimulation of the immune response caused changes in inflammatory markers, an increase in cholesterol and triglyceride levels and a decrease in high-density lipoprotein (HDL), contributing to the development of dyslipidemia, another well-known cardiovascular risk factor. The consequences of these processes are atherosclerotic disease, which eventually paves the way to a prothrombotic status and potentially to IHD [55].

In heart disease caused by bacterial infection, IE is one of the most common causes in young age groups (with an average of 35 years of age), basically because a high percentage of patients have some congenital valvular disease predisposing them to the development of endocarditis or other conditions associated with IE, such as states of immunosuppression, e.g., cirrhosis, cancer or organ transplant [56]. 

Infective endocarditis can be classified into bacterial endocarditis and nonbacterial endocarditis, generally caused by viruses, fungi and other microbiological agents that make the diagnosis difficult, particularly during the early stages of the disease. In earlier times, *Streptococcus* spp. were the most common and predominant cause of native valvular IE. However, increasing patient age and increasing nosocomial infections have led to major changes in the nature of IE infection [57].

As expected, in a cohort of nearly 1200 cases of infective endocarditis, the plurality of cases was attributable to *Staphylococcus aureus (S. aureus)* (32%), with 18% *S. viridans*, 11% *Enterococcus* spp. and 11% coagulase-negative *Staphylococcus* spp. 

A cohort study conducted in 16 countries by Fowler et al. enrolled 1779 patients with IE as defined by the Duke criteria.

The results demonstrated that *Staphylococcus aureus* was the most common pathogen among the 1779 cases, supporting the hypothesis that *S. aureus* is the leading cause of IE in many different regions of the world [58].

As discussed above, group A Streptococcus (GAS) is a gram-positive bacterium that can be divided based on its M proteins into more than 100 M serotypes [59]. Complications that can arise after infection with *Streptococcus* spp. include retropharyngeal abscess as well as complications such as acute rheumatic fever and kidney infections. 

GAS infections have remained susceptible to penicillin, so it currently remains the first-line antibiotic therapy, although higher concentrations of penicillin are currently needed to inhibit the growth of some other streptococcal bacteria, including *Streptococcus pneumoniae* [60].

Heart diseases can also be caused by a direct cytopathic effect of viruses or a relentless pathological immune response triggered by viral infection.

Diseases such as heart abnormalities, congenital infection and transplacental infection during pregnancy are usually caused by CMV and the rubella virus [61]. 

Infections caused by these viruses are currently one of the most common causes of abortion in developing countries, and human cytomegalovirus (HCMV) infection is strongly associated with coronary heart disease. 

In support of this hypothesis, several studies (in serology as well as molecular biology) have documented that HCMV-infected endothelial cells play a crucial role in the progression of aortic stenosis. A high percentage (52%) of AS lesions gave a HCMV antigen-positive result, compared with non-AS tissues (29% positive), confirming that HCMV is a starting factor of AS development [62,63].

Additionally, a study involving over 14,000 patients showed that those who had undergone cardiopulmonary resuscitation gave a HCMV-positive result and reported a significantly higher mortality rate than the patients negative for HCMV antibodies. Finally, this study demonstrated a correlation between HCMV and heart disease [64].

Additionally, Coxsackie B virus infection (CVB) has been shown to be one cause of primary myocardial disease and to be involved in the development of chronic cardiovascular diseases [65].

In recent years, the concept of circulating microbiota has been recognized as another key player in health and disease. Even if the presence of microbes in the blood has long been known, growing evidence is providing new insights into the role of the circulating microbiota in the pathogenesis of several diseases, such as cardiovascular diseases.

In addition, the theory of a healthy blood microbiota has been strengthened by accumulating evidence of microbes in healthy human blood, as ascertained by measurements of bacterial metabolism, microscopic observation, blood culture, quantitative PCR and next generation 16S rRNA gene sequencing (NGS and shotgun metagenome sequencing) [66].

Microbial DNA is physiologically localized in cellular components, and the blood of healthy subjects is usually sterile, with no bacterial growth, in contrast to the blood of sick people. 

In a recent study regarding blood microbiota dysbiosis associated with diabetic disease, no bacterial growth was found in the blood plasma of healthy individuals, in contrast to the blood of diseased patients. Moreover, a big difference in bacterial diversity was observed between the healthy gut microbiota and the healthy blood microbiota. The gut is predominantly inhabited by two major bacterial phyla, namely *Firmicutes* and *Bacteroidetes*, whereas peripheral blood from healthy donors is mostly dominated by the *Proteobacteria* phylum. [67]

In addition, another study by Amar et al. observed that blood microbiota dysbiosis was significantly associated with the onset of cardiovascular events. In particular, during CVD the blood microbiota undergoes a complete transformation in microbial diversity, with a dominance of intestinal bacteria *Firmicutes* and *Bacteroidetes*, as a result of gut barrier disruption, underlining how blood microbiota dysbiosis is significantly associated with the onset of cardiovascular events [68].

In conclusion, the changes demonstrated in microbiota composition in cases of CVD and also the role of the blood microbiota, together with circulating microbial metabolites, can be considered as potential tools for clinical practice, both to monitor the clinical status of patients and to evaluate new therapeutic opportunities for improved cardiovascular health. 

## 3. Pharmacomicrobiomics’ Role in Personalized Medicine

Pharmacomicrobiomics is a new branch that proposes to describe the influence of the microbiome on xenobiotic action, appropriately studying the interactions between the host, gut microbiota and drug action, as shown in Figure 1. 

In the last few years, the intestinal microbiome has been considered as a “second genome” that plays an important role in both healthy status and drug response, and antibiotic administration, fecal microbial transplantation and probiotic treatment have been identified as good strategies for the shaping of the microbiota [69]. 

The advancement in microbial genomics from culture-based to culture-independent methodologies, such as metagenomics or shotgun sequencing of microbial and viral communities, has led to the identification of the gut microbiome’s genetic role linked to disease development or altered therapeutic response [70].

Pharmacomicrobiomic studies have shown that gut microorganisms and their enzyme products can affect the bioavailability, clinical efficacy and toxicity of a variety of drugs by direct and indirect mechanisms. 

The advent of this new discipline promises to simplify the microbiome-based, personalized medicine approaches in several diseases [71,72]. Nowadays, it can be defined as a proactive, well-coordinated and well-proven structural strategy for effective healthcare which works best with a network of electronic health records that combine clinical and molecular data to make the best treatment options possible, allowing for the development of focused care for patients who do not respond to medications as expected. [73].

The first recognized role of the microbiota as a modulator of drug activity was in the 1930s with prontosil, an antibacterial chemotherapy molecule developed by a German research group as a precursor to the microbial-derived antibiotic sulfanilamide. In particular, it emerged that prontosil itself was not endowed with antibacterial activity but, once taken by a mouse, underwent a metabolic degradation which led to the synthesis of an antibiotic molecule, sulfonylamide [69,74].

Subsequently, sulfasalazine, a drug used for the treatment of ulcerative colitis, Crohn’s disease and rheumatoid arthritis, confirmed the important role of the microbiome in drug activity. This anti-inflammatory molecule undergoes a reductive metabolism by intestinal bacteria to be converted into sulfapyridine and 5-aminosalicylic acid, a non-steroidal anti-inflammatory drug available for systemic absorption [75].

Considering the high variability in drug response and the microbiota’s complexity, it would be necessary to develop a systematic method to obtain the most effective therapeutic results in the host. 

A good personalized medicine approach was tested by an Israeli study group that monitored week-long glucose levels in 800 healthy and pre-diabetic people, measuring responses to 46,898 meals. Variables such as microbiota composition, blood tests, food diary, genotyping and anthropometrics were recorded using machine learning algorithms. The obtained results were very interesting, since they found a high variability in the responses to identical meals between the participants, with significant blood glucose variability and consistent taxa changes in most participants, suggesting a role for the microbiome in postprandial variability [76].

Assuming that the gut microbiome played a central role in the pathogenesis and progression of inflammatory bowel disease (IBD), a recent prospective study demonstrated that the evaluation of the gut microbiome can also predict responses to IBD therapy. In detail, a cohort of patients with IBD was enrolled for gut-selective anti-integrin therapy with vedolizumab.

Vedolizumab is a recombinant humanized immunoglobulin G1 (IgG1) directed against the human LPAM (lymphocyte Peyer’s patch adhesion molecule 1) with immunomodulating, anti-inflammatory and potential antineoplastic activities. Disease course and stool metagenomes were assessed at baseline and after treatment, starting at 14, 30 and 54 weeks.

Using a network algorithm that incorporated both microbiome and clinical data of patients, they highlighted the associations between the basic taxonomic GM composition and functional abundance as well as clinical remission at 14 weeks, demonstrating the usefulness of the model predictors in clinical remission and hypothesizing that monitoring early changes in the microbiota may be a valid marker in IBD treatment [77].

### Pharmacomicrobiomics Focuses on Cardiovascular Diseases

There is growing interest in pharmacomicrobiomics and the microbiome’s role in CVD. 

Digoxin, in particular, represents a typical example of how the microbiome can influence a drug’s effect on the heart. Digoxin, also called digitalis, helps an injured or weakened heart to pump more efficiently and is very useful for maintaining clinical stability and exercise capacity [78]. Despite this, when digoxin is used in low-flow congestive heart failure, it does not work in one out of ten patients, because it is inactivated by certain specific bacterial strains present in the intestine, underlining the importance of pharmacology from a human and a microbial point of view. The reasons for this deactivation were later attributed to the conversion of digoxin into an inactive form, dihydrodigoxin, by the gut Actinobacterium *Eggerthella lenta* (E. *lenta)*, previously named *Eubacterium lentum*. The process is shown in Figure 2 [79,80,81,82].

In a very interesting study conducted by Haiser et al., the authors identified an operon encoding cytochrome in the common intestinal bacterium *E. lenta* that was transcriptionally activated by the cardiac drug digoxin. 

Using RNA-seq to identify differentially expressed *E. lenta* genes, they identified this operon, the cardiac glycoside reductase (cgr), which was upregulated in the presence of digoxin. Therefore, these genes represent a predictive microbial biomarker for digoxin inactivation. To support this, a quantitative PCR of bacterial DNA isolated from the stool of 20 healthy volunteers was performed, confirming a significant correlation between the abundance of the cgr operon, normalized to the level of *E. lenta* 16S rDNA, and digoxin inactivation. Moreover, they considered digoxin signaling in eukaryotic systems, pondering the possibility that endogenous, digoxin-like molecules may have selected the deactivation and determining whether digoxin inactivation in vivo could be controlled by rational dietary interventions. This last point raised great interest, since it was already known that the growth of *E. lenta* requires the amino acid arginine, which, while improving growth, also inhibits the inactivation of digoxin. As a result, higher arginine levels from dietary or microbial sources could be used to inhibit this unfavorable microbial activity [83,84]. Therefore, they performed an in vivo experiment to test whether high levels of arginine from food could be useful in preventing this undesirable microbial activity. Mice were fed on two different diets, one completely lacking in protein and the other providing 20% kcal from protein. This in vivo experiment revealed that increasing dietary protein considerably increased serum and urinary digoxin levels, and this happened only in mice colonized with the strain that reduces digoxin [83].

Studying the mechanisms that regulate the functions of cells in the heart will allow the identification of novel molecular targets for pharmacological intervention and will assist the future development of therapeutic strategies for managing cardiovascular disorders [85]. In relation to what was previously stated, the potential role of TMAO, already studied in several diseases, has been considered for AS development as a therapeutic target that can be regulated by keeping plasma levels under control [86]. TMAO, in fact, can be targeted through diet, dietary supplements and lifestyle interventions. The use of antibiotics, prebiotics, probiotics and some specific natural molecules can substantially reduce the levels of TMAO by remodeling or modifying the intestinal microbiota. In addition, some plant sterol esters (PSE) have been shown to significantly dampen microbial production of TMAO, attenuating cholesterol accumulation and nearly abolishing atherogenesis [87]. In light of these studies, the idea of personalized medicine is taking hold. This new medical field provides individualized clinical decisions and procedures specific for each patient. Genetic information is considered as a means to be used in preventive and therapeutic strategies, allowing the treating physician to provide better therapy, in terms of efficacy, safety and effectiveness of the treatment, to their patients [88].

## 4. Concluding Remarks and Future Perspectives

There is much evidence linking microbiotas to cardiovascular disease therapy, suggesting that these two factors can modulate each other. 

Consequently, the study of how gut microorganisms can impact on drug toxicity and metabolism could be very useful to improve the targeted action of drugs and open a new approach for personalized medicine [89]. 

Nowadays, to improve therapeutic outcomes and alleviate adverse drug effects, a number of approaches to selectively manipulate the microbiota have been suggested, including the administration of probiotics and prebiotics to support the conventional treatments [90]. Therefore, pharmacomicrobiomics represents an additional factor for improving the innovative concept of personalized medicine, which aims to prevent, treat and cure several diseases, including CVDs, and to promote longitudinal wellness in patients suffering from crippling diseases such as cardiomyopathies. In the wake of these advances, there has been an exponential development of targeted drugs that allow clinicians to make choices guided by the personal characteristics of each patient, avoiding drug inactivation and side effects, i.e., precision medicine. This new branch is designed to incorporate an integrated and multidisciplinary approach which combines different types of data to create a healthcare model tailored to the individual patient. Moreover, new approaches and techniques have favored the division of the population into many subgroups with different characteristics and needs which require different and specific treatments [91]. It, therefore, emerges that pharmacomicrobiomics is an innovative approach to the administration of personalized medicine, improving both drug efficacy and safety for patients. Pharmacomicrobiomics is particularly powerful, as it is sensitive to both genetic and environmental factors, such as diet, drug intake and, most importantly, the microbiome. The characterization of microbiotas has become more and more important over the years. We could predict that pharmacomicrobiomics will be equally important in the coming decades and will be valuable in itself and complementary to microbiome evaluation.

Similarly, phage therapy could represent a promising alternative in the treatment of certain bacterial infections, multiresistant or not, playing an important role in promoting and contrasting drug inactivation pathways. In fact, current research on the use of phages and their lytic proteins against multidrug-resistant bacterial infections highlights their potential use as a valid supplement to antibiotic therapies [92].

To conclude, the consideration of individual patients as a complex holobiont (in other words, the host and its associated communities of microorganisms) will help to conceive new approaches for predictive, preventive and personalized medicine; to overcome many challenges, by contributing to the advancement of medical practice, especially in CVD fields, and transforming the future of personalized healthcare; and could be the right trigger for the development of new therapeutic interventions.

## Figures and Tables

**Figure 1 biomedicines-09-01338-f001:**
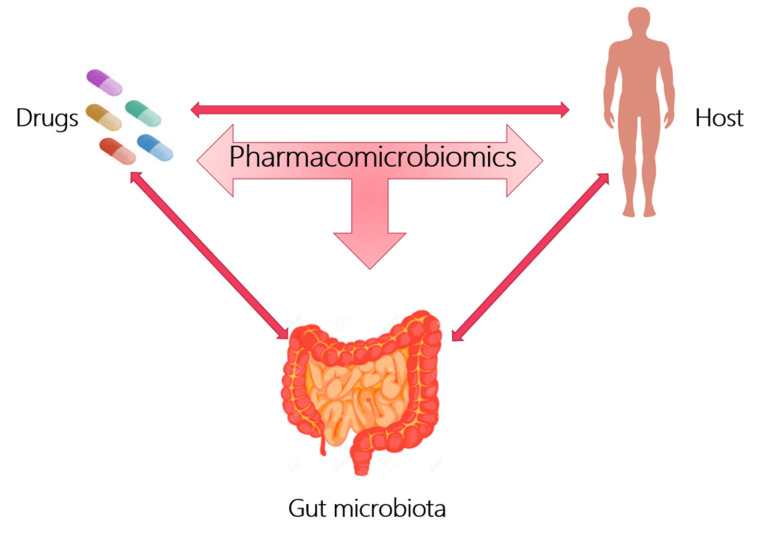
Pharmacomicrobiomics studies the interaction between drugs, host and gut microbiota.

**Figure 2 biomedicines-09-01338-f002:**
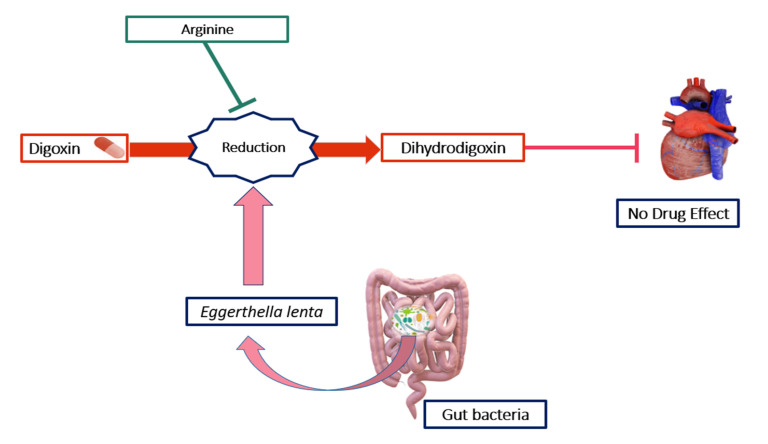
Digoxin is a typical example of how the gut microbiota can influence drug metabolism by deactivating its effect. The reason for this deactivation is attributed to the conversion of digoxin into an inactive form, dihydrodigoxin, by the gut bacterium *Eggerthella lenta*. On the other hand, arginine prevents the reduction of digoxin to dihydrodigoxin, making the drug work. This could also explain interindividual variations in drug functionality.

## Data Availability

Not applicable.

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
