# Peer review of "Cardiovascular Diseases and Pharmacomicrobiomics: A Perspective on Possible Treatment Relevance"

_biomedicines, 2021, doi:10.3390/biomedicines9101338_

Round 1
Reviewer 1 Report
The main weak points of the presented review are:
- Lack of a critical review and discussion of up to date experimental/research articles – out of 87 reference positions only 14 papers (16%) describe experimental works and were published in the last 10 years; five experimental references were from the last century
- Lack of an original Authors’ contribution in the presented theory – out of 87 reference positions as many as 61 are review papers, and almost all statements concerning human blood microbiome, human gut microbiome or relationship between them and human health are supported by literature. Of course, it is not disqualifying when scientists have the same point of view on a certain issue. The problem is that the presented work is more a review of reviews than an original review, which I understand as an original selection of research papers, discussed and commented by Authors in order to present the state-of-the-art in the field and/or to present Authors’ point of view on a certain problem, theory ect. Here, as few as 30% of all references are experimental/research articles.
- Inappropriate composition of the manuscript – most of the attention is paid to the cardiovascular diseases and human blood microbiome, whereas pharmacomicrobiomics (the main topic of the review) takes only a small part of the manuscript and is supported by only four(!) experimental articles, three of which were published in 1976, 1983 and 1999.
A strong point of the manuscript – it is an excellent source of the newest and high quality review papers on cardiovascular diseases, human blood microbiome, human gut microbiome and pharmacomicrobiomics.
Minor remarks that do not impact on the final manuscript recommendation:
The manuscript must be spell checked and grammar checked by a native speaker.
Line 18 please correct – “could be potentially correlate with the CVDs” – “could potentially correlate with the CVDs” or “could be potentially correlated with the CVDs”
Line 60 please correct “can be quantified and”
Line 87—88 please correct “has been identified”
Line 180-182 please correct the sentence syntax, in current form the sentence is unclear
Please merge text in lines 203-210 into one paragraph
Please merge text in lines 211-218 into one paragraph
Please merge text in lines 219-237 into one paragraph
Line 248 “Microbial DNA in healthy people” – it is confusing, please reformulate
Line 256-257 please revise or omit this sentence. I have not found such a statement in the given reference.
Line 312 please correct “interaction”
Figure 2 – please modify the figure. Now, it can be read that gut bacteria produce digoxin, digoxin is modified by Eggethella lenta to dihydrodigoxin. The impact of arginine on E. lenta activity should be included in the figure.
Please merge text in lines 369-381 into one paragraph
Author Response
The authors would like to thank the Biomedicines reviewers for their specific and helpful comments about our Review for the Special issue “CARDIOVASCULAR DISEASES: COULD THE PHAR-MACOMICROBIOMICS GIVE SUPPORT FOR TREATMENTS?”.
The manuscript has been improved according with the comments suggested.
- Point 1: Lack of a critical review and discussion of up to date experimental/research articles – out of 87 reference positions only 14 papers (16%) describe experimental works and were published in the last 10 years; five experimental references were from the last century
Reply 1: Following the right suggestions of reviewer, we have rewritten many manuscript parts. In detail, we have added the original studies to replace the reviews, specifically: in the line 55, references 9 and 10 were added to replace two reviews; in the lines 60 and 64, references 12 and 13 (two articles) were added; respectively in line 82 (reference 18) and in line 91 (references 19 and 20) were added to replace a dated work; line 292 was added an article (reference 67) to replace a review; line 298 (reference 68) was added an article; line 383 ( references 79) was added an article to replace a dated work; line 401 (reference 81) was added a clinical trial to replace a dated work.
Moreover, in agreement with the reviewer suggestions, we highlighted our critical point of view about the cited manuscripts, including e.g..: lines 66-71 on which we discuss about TMAO role giving our idea; lines 100-109 we tried to give our critical point of view about the topic discussed above; lines 303-307 we report our conclusive considerations regarding the use of some parameters, such like the circulating microbial metabolites as valid predictive tools in cardiovascular diseases.
- Point 2: Lack of an original Authors’ contribution in the presented theory – out of 87 reference positions as many as 61 are review papers, and almost all statements concerning human blood microbiome, human gut microbiome or relationship between them and human health are supported by literature. Of course, it is not disqualifying when scientists have the same point of view on a certain issue. The problem is that the presented work is more a review of reviews than an original review, which I understand as an original selection of research papers, discussed and commented by Authors in order to present the state-of-the-art in the field and/or to present Authors’ point of view on a certain problem, theory etc. Here, as few as 30% of all references are experimental/research articles.
Reply 2: We thank the reviewer for right and constructive suggestions. As previously reported, we have tried to qualitatively improve your manuscript bringing out our considerations and our point of view regarding the cited original papers. In addition we added more research articles than the first version submitted.
- Point 3: Inappropriate composition of the manuscript – most of the attention is paid to the cardiovascular diseases and human blood microbiome, whereas pharmacomicrobiomics (the main topic of the review) takes only a small part of the manuscript and is supported by only four(!) experimental articles, three of which were published in 1976, 1983 and 1999.
Reply 3:
We thank the reviewer for the adequate suggestion and the possibility to explain your paper idea. The purpose of our work was to focus in the pharmacomicrobiomics as a new strategy to support the current treatments of cardiovascular pathologies that often can be invalidate by the microbiota interference, making the therapy of patients more personalized and effective. Being this a more innovative topic, there are currently not many original articles that address this view, and that deal with the connection here proposed between pharmacomicrobiomics and cardiovascular diseases. This is why our idea to propose this Review talking about pharmacomicrobiomics, of personalized medicine and of the huge potential that these branches can offer to patients, but correlating them to cardiovascular diseases that are increasingly widespread. In addition, we have added original articles and recent reviews to support the topic of pharmacomicrobiomics. Moreover, following the reviewer suggestions, in line 91, references 19 and 20 were added to replace the dated work of the year 1999; line 383, references 79 was added an article to replace the dated work of the year 1983; line 401, reference 81 was added a clinical trial to replace the dated work of the year 1976.
A strong point of the manuscript – it is an excellent source of the newest and high quality review papers on cardiovascular diseases, human blood microbiome, human gut microbiome and pharmacomicrobiomics.
Reply:
We thank the reviewer for the positive judgment.
Minor remarks that do not impact on the final manuscript recommendation:
Question (Q) - Response (R)
Q: The manuscript must be spell checked and grammar checked by a native speaker.
R: As suggested, a native speaker. has checked the English grammar and corrected the oversights
Q: Line 18 please correct – “could be potentially correlate with the CVDs” – “could potentially correlate with the CVDs” or “could be potentially correlated with the CVDs”
R: In agreement with the suggestion, the sentence has been corrected
Q: Line 60 please correct “can be quantified and”
R: Following the suggestion, the sentence has been corrected
Q: Line 87—88 please correct “has been identified”
R: In agreement with the reviewer comment, the sentence has been corrected
Q: Line 180-182 please correct the sentence syntax, in current form the sentence is unclear
R: Following the suggestion, the sentence syntax has been corrected
Q: Please merge text in lines 203-210 into one paragraph
R: As suggested, the text has been merged
Q: Please merge text in lines 211-218 into one paragraph
R: Following the suggestion, the text has been merged
Q: Please merge text in lines 219-237 into one paragraph
R: Following the suggestion, the text has been merged
Q: Line 248 “Microbial DNA in healthy people” – it is confusing, please reformulate
R: As suggested, the sentence syntax has been corrected
Q: Line 256-257 please revise or omit this sentence. I have not found such a statement in the given reference.
R: Following the suggestion, the sentence syntax has been corrected
Q: Line 312 please correct “interaction”
R: Following the suggestion, the word has been corrected
Q: Figure 2 – please modify the figure. Now, it can be read that gut bacteria produce digoxin, digoxin is modified by Eggerthella lenta to dihydrodigoxin. The impact of arginine on E. lenta activity should be included in the figure.
R: In agreement with the reviewer, Figure 2 and the Caption were both modified and the role of arginine has also been added
Q: Please merge text in lines 369-381 into one paragraph
R: Following the suggestion, the text has been merged
Reviewer 2 Report
The present submitted review deals with the study of heart diseases highlighting how the microbiota plays a relevant role in their development.
Finally, the authors investigate pharmacomicrobiomics, a new interesting branch, and the microbiota role in modulating the response to drugs to improve their effectiveness and targeted action.
I find that the scientific problem discussed in this Review is original and of scientific interest and the reported literature references and discussion are updated, huge and complete.
I ask the authors to check carefully some English grammar, syntax and typing errors along the paper.
All the acronyms must be specified the first time they are used.
Author Response
The present submitted review deals with the study of heart diseases highlighting how the microbiota plays a relevant role in their development.
Finally, the authors investigate pharmacomicrobiomics, a new interesting branch, and the microbiota role in modulating the response to drugs to improve their effectiveness and targeted action.
I find that the scientific problem discussed in this Review is original and of scientific interest and the reported literature references and discussion are updated, huge and complete.
Q: I ask the authors to check carefully some English grammar, syntax and typing errors along the paper.
R: As suggested, a native speaker has checked the English grammar and corrected the oversights
Q: All the acronyms must be specified the first time they are used.
R: Following the reviewer suggestion we have specified all the acronyms at the first use
We hope our modification make the Review suitable for publication.
Yours sincerely.
Round 2
Reviewer 1 Report
The Authors tried to address the indicated main weak points (they included more experimental references and added own comments to the text) and updated the literature in the section dedicated to pharmacomicrobiomics. Unfortunately, in my opinion, the changes made in the text and in the reference section (replacement about ten review articles with experimental ones) are not sufficient to accept the manuscript for publication. This manuscript it still more a source of review papers rather than a standalone review article.
It is disappointing that Authors did not address the minor remarks:
- There are still single-sentence paragraphs, moreover, the Authors added new ones (e.g. l. 456-460)
- The Figure 2 is even more misleading than before – a scheme showing that gut bacteria produce digoxin (what is not true) has not been corrected but duplicated
I want to underline, that the minor remarks had no influence on the final recommendation for the manuscript (rejection).
Author Response
Point 1: There are still single-sentence paragraphs, moreover, the Authors added new ones (e.g. l. 456-460)
Reply 1: In accordance with the suggestions, the text has been merged in a single paragraph (e.g. lines 425-429; 449-456; 456-460; 460-463).
Point 2: The Figure 2 is even more misleading than before – a scheme showing that gut bacteria produce digoxin (what is not true) has not been corrected but duplicated
Reply 2: According to your suggestions, we have modified the Figure 2 and the caption, in order to better clarify the digoxin inactivation process.